# Isavuconazole and Amphotericin B Synergic Antifungal Activity: In Vitro Evaluation on Pulmonary Aspergillosis Molds Isolates

**DOI:** 10.3390/antibiotics13111005

**Published:** 2024-10-25

**Authors:** Maddalena Calvo, Flavio Lauricella, Anna Maurizia Mellini, Guido Scalia, Laura Trovato

**Affiliations:** 1Department of Biomedical and Biotechnological Sciences, University of Catania, 95123 Catania, Italy; maddalenacalvo@gmail.com (M.C.); flavio.lauricella@gmail.com (F.L.); annamellini17@gmail.com (A.M.M.); lido@unict.it (G.S.); 2U.O.C. Laboratory Analysis Unit, A.O.U. Policlinico “G. Rodolico-San Marco” Catania, 95123 Catania, Italy

**Keywords:** isavuconazole, amphotericin B, in vitro synergy, *Aspergillus* spp., pulmonary aspergillosis

## Abstract

**Background/Objectives.** Pulmonary aspergillosis is a severe respiratory infection caused by *Aspergillus* spp., whose resistance profiles and invasive attitude complicate therapeutical strategies. Several aspergillosis cases emerged as superinfections during the SARS-CoV-2 pandemic when isavuconazole and amphotericin B became essential antifungal alternatives. The main purpose of the present study was to investigate a possible synergic activity between these molecules against *Aspergillus* spp. isolated from respiratory samples. **Methods**. The gradient test method detected isavuconazole and amphotericin B MIC values, prompting an arrangement of their combination into an R.P.M.I. agar medium. According to Liofilchem s.r.l. instructions, the FIC index was used to establish synergy, additivity, indifference, or antagonism. **Results**. Among 36 *Aspergillus* spp. isolates, only *A. fumigatus* strains showed both synergy and additivity episodes. *A. niger* reported the highest antagonism percentage, while *A. terreus* revealed several indifference episodes. **Conclusions**. Isavuconazole and amphotericin B remain fundamental therapeutical alternatives, including a possible synergic effect against *A. fumigatus*. On the basis of this species-related difference, further studies will be essential to investigate different antifungal drug combinations against filamentous fungi isolates.

## 1. Introduction

Pulmonary aspergillosis is a critical respiratory infection caused by *Aspergillus* spp. This genus includes ubiquitous molds, which are commonly diffused in the external environment, potentially impacting the human respiratory tract [1]. *Aspergillus* spp. may cause invasive pulmonary aspergillosis, an acute bronchial and pulmonary blood vessel aggression by the molds’ hyphae. Immunocompromised patients represent the main target due to their intense risk factors such as neutropenia, bone marrow or solid organ transplantation, and prolonged chemotherapy [1,2]. Specifically, the level of neutropenia is heavily related to increased invasive pulmonary aspergillosis risk. Otherwise, invasive pulmonary aspergillosis cases emerged among intensive care unit (ICU) patients after mechanical ventilation usage due to respiratory failure. This condition recently characterized patients affected by severe influenza syndrome or COVID-19 disease who initially show immunocompetence [3]. Progressively, SARS-CoV-2 infections cause severe immunomodulation and white blood cell depletion, leading to a significant immunological impairment. Furthermore, this viral infection requires corticosteroid administration, consequently impacting the immune system. On that premise, superinfections may become extremely probable among similar patients. *Aspergillus* spp. respiratory infections recently emerged as a frequent COVID-19 complication, suggesting the urgency of optimizing diagnostic and clinical management [3].

Despite the recent advancement in diagnostic microbiology, pulmonary aspergillosis still represents a significant diagnostic challenge [1,2]. Luckily, galactomannan antigen research on serum and bronchoalveolar lavages provides fundamental diagnostic added value to the diagnostic workflow. Moreover, molecular technologies investigating the presence and potential resistance markers of *Aspergillus* spp. have recently been integrated into pulmonary aspergillosis diagnosis guidelines [4]. According to epidemiological reports, *Aspergillus fumigatus* represents the most common aspergillosis cause, followed by *Aspergillus flavus*, *Aspergillus terreus*, and *Aspergillus niger*. Few aspergillosis cases (less than 3%) may be related to *Aspergillus nidulans*, *Aspergillus calidoustus*, *Aspergillus glaucus*, *Aspergillus versicolor*, *Aspergillus lentulus*, and *Aspergillus udagawae* [3].

Pulmonary aspergillosis also shows therapeutical defiance due to potential antifungal resistance or patients’ immunological impairments. Guidelines currently include polyenes, triazoles, and echinocandins for aspergillosis treatment. Furthermore, ultimate antifungal frontiers such as olorofilm and fosmanogepix are under investigation for the same therapeutical application, especially in the case of azole resistance [3]. Liposomal amphotericin B represents the most important polyenes member and its clinical usage regards invasive pulmonary aspergillosis as a secondary therapeutical choice after the triazole (first choice only in the case of azole resistance documented by antifungal susceptibility testing) [3]. Echinocandins (caspofungin, anidulafungin, and micafungin) are included among the therapeutical alternatives, inhibiting the β-(1,3)-D-glucan synthesis within the fungal cell wall. However, their effectiveness against *Aspergillus* spp. is lower than triazoles [5]. Amphotericin B interacts with the fungal ergosterol membrane, leading to altered permeability and cell mortality. However, its usage should be strictly monitored due to potential renal impairment [5]. Isavuconazole, itraconazole, and voriconazole are the main choices of triazole in the case of invasive pulmonary aspergillosis, altering ergosterol synthesis within the cell membrane. Specifically, isavuconazole is a novel triazole associated with less hepatotoxicity and more pharmacokinetic stability, which does not need strict therapeutic drug monitoring (TDM) compared to voriconazole [3,6]. Furthermore, this ultimate triazole frontier is not affected by some common azole resistance mechanisms. For instance, the G54 mutations only impact voriconazole antifungal activity [7]. Isavuconazole has been extensively used in CAPA treatment in recent years [8,9]. Additionally, both isavuconazole and liposomal amphotericin B are essential therapeutical options for invasive pulmonary aspergillosis in the ICU [8]. 

Previous studies reported in vitro susceptibility data on isavuconazole activity on *Candida* spp., rare yeasts, and *Aspergillus* spp. [10,11]. Moreover, published data recently documented the potential synergic activity of isavuconazole and amphotericin B against *Candida* spp. yeasts and some filamentous fungi such as *Mucorales* [12,13]. Unfortunately, databases do not report in vitro susceptibility data about the same synergism against *Aspergillus* spp. Herein, we propose an experimental in vitro evaluation of the synergic isavuconazole and amphotericin B activity on *Aspergillus* spp. isolated from respiratory samples of pulmonary aspergillosis patients. The main study purpose was to enrich the literature data about this topic, also aiming to demonstrate possible variability in synergic antifungal activity depending on the isolated *Aspergillus* spp. species. 

## 2. Results

The experimental protocol globally collected 36 *Aspergillus* spp. isolates from various respiratory samples. Specifically, the study used strains derived from the intensive care unit (15), the hematology ward (9), the infectious diseases ward (5), the pneumology unit (3), the emergency room (2), and the surgery clinic (2). The microbiological analysis identified *A. fumigatus* (14), *A. flavus* (8), *A. niger* (7), and *A. terreus* (7). Figure 1 describes these species’ incidence within the analyzed hospital units. 

Remarkably, most of the isolates belonged to the *A. fumigatus* (38.8%) and *A. flavus* (22.2) species, while slightly lower incidence rates were generally reported for *A. niger* (19.4%) and *A. terreus* (19.4%). Moreover, *A. fumigatus* was the only species isolated from infectious disease patients, while *A. niger* was primarily seen within the intensive care unit. 

Table 1 reports all the details about isavuconazole and amphotericin B MIC values for all the investigated isolates. Additionally, Table 2 summarizes the isavuconazole and amphotericin B MIC ranges obtained from the MIC Strip test and EUCAST broth microdilution. Furthermore, the same table reports a susceptible/resistant (S/R) categorization according to clinical breakpoints or an epidemiological cut-off. Finally, the authors documented the percentage of isolates whose MIC values overcame the E-COFF value for both the tested antifungal drugs. Interestingly, all the *Aspergillus* spp. strains revealed in vitro isavuconazole susceptibility. As regards amphotericin B, *A. fumigatus* was the only species to report a complete susceptibility rate. *A. niger* and *A. flavus* revealed one amphotericin B-resistant strain for each species. Despite the known inherent amphotericin B resistance, *A. terreus* reported a susceptible result for one strain. The MIC Strip method MIC values were comparable to the EUCAST broth dilution ones, revealing a total coherence in S/R categorization. 

Synergy evaluation revealed some differences depending on the tested species. *A. fumigatus* was the only species reporting synergy episodes between amphotericin B and isavuconazole. Furthermore, the same species achieved the highest additivity and the lowest antagonism rates. Consequently, it was the only species documenting statistical significance for the synergy investigation. Additionally, *A. flavus* showed the lowest additivity rate, while *A. niger* reported the highest antagonism percentage. Finally, *A. terreus* documented the highest indifference rate. Table 3 summarizes the synergy evaluation results for all the identified species, along with the corresponding statistical significance. 

## 3. Discussion

Pulmonary aspergillosis constitutes a relevant global health issue, especially among immunocompromised and intensive-care patients. Severe viral respiratory infections may significantly compromise those with this condition or predispose individuals to its subsequent development. SARS-CoV-2 infections were often associated with pulmonary aspergillosis cases in recent years, leading clinicians to establish specific CAPA management guidelines [15,16]. We decided to focus our attention on pulmonary aspergillosis cases due to its possible diagnostic underestimation, especially in the case of contextual or previous viral respiratory infections. For instance, in most COVID-19 cases, steroid-based treatments impair the patient’s immune system, leading to potential fungal superinfections, with *Aspergillus* spp. counting among the possible etiological agents [17,18,19,20]. The CAPA episodes may be managed through isavuconazole and amphotericin B, considering potential azole resistance cases or amphotericin B nephrotoxicity [16,21]. These considerations significantly complicate the therapeutic management of pulmonary aspergillosis, leading us to further investigate in vitro *Aspergillus* spp. susceptibility profiles. Specifically, there is a lack of published data investigating antifungal molecules’ synergic activity against *Aspergillus* spp. On these premises, our initial hypothesis was to test amphotericin B and isavuconazole combinations against different *Aspergillus* species isolated from respiratory samples of COVID-19 and pulmonary aspergillosis patients.

First, the study reported *A. fumigatus* as the most isolated species, followed by *A. flavus* and *A. niger*. Otherwise, *A terreus* was the less common isolate. These data approximately match the previously documented epidemiology within the southern Italy area [22]. Remarkably, *A. fumigatus* is also the most virulent *Aspergillus* species due to its capability to extensively generate hyphae invading host cells and human tissues [23]. 

We investigated potential synergy through the MIC Strip method to furnish easy-to-handle laboratory data. This method previously demonstrated high agreement rates to broth microdilution [11,12], encouraging its application in susceptibility testing within the laboratory diagnostic routine. As regards synergy testing methods, the checkerboard represents a gold standard technique. However, few literature data have been published about the use of this method on *Aspergillus* spp. isolates [24]. We decided to apply the MIC Strip method to investigate in vitro isavuconazole and amphotericin B synergy to gather quick susceptibility data within the laboratory diagnostic routine. Our results demonstrated a significant difference in synergy outcomes depending on the analyzed *Aspergillus* species. In our opinion, this evidence enhances the importance of always performing species identification in the laboratory diagnostic routine. Different species may reveal specific characteristics of the antifungal susceptibility profile. Notably, the common and virulent *A. fumigatus* was the only species to report synergy and the most frequent species to reveal additivity cases. Similar data may encourage further studies about the combination of isavuconazole and amphotericin B against this species. These results were probably derived from amphotericin B alone, the *A. fumigatus* MIC results of which were the lowest during our analysis. We hypothesized that their low value significantly impacted the definitive FIC index calculation after the synergy testing. 

On the other hand, several antagonism or indifference episodes belonged to *A. niger*, *A. terreus*, and *A. flavus*, which never reported synergy cases during the study. Additionally, these species rarely revealed additivity. We hypothesized that a combination of echinocandin and isavuconazole could overcome these episodes due to the capability of both molecules to target the same cell component. However, preliminary studies have already investigated this hypothesis, demonstrating that frequent indifference episodes may occur for all the main *Aspergillus* species [25]. 

As a consequence, future studies should be planned including a larger isolate spectrum investigating both amphotericin B/isavuconazole and amphotericin B/echinocandin combinations. 

Notably, most of our cases confirmed low MIC values both for amphotericin B and isavuconazole, highlighting their essential role within antifungal therapeutical plans [26,27]. Scientific evidence demonstrated the importance of amphotericin B in treating triazole-resistant aspergillosis. These cases reported significant amphotericin B effectiveness against *Aspergillus* spp., except for the frequent resistance episodes of *A. terreus*. Amphotericin B resistance or prolonged administration may be avoided through isavuconazole usage [28]. This drug confirmed in vitro effectiveness as a broad-spectrum triazole during our experimental evaluation, enforcing the literature data about its clinical efficacy [29]. However, some species-related features emerged. One *A. flavus* and one *A. niger* strain showed a high amphotericin B MIC value, resembling previously published resistance episodes [30,31]. Finally, one *A. terreus* isolate revealed a low amphotericin MIC value. Despite this species’ well-known inherent amphotericin B resistance, the literature data documented several antifungal tolerance episodes. These cases report in vitro susceptibility along with in vivo therapeutical failure due to the species’ capability to adapt its growth to different antifungal drug concentrations [32]. 

## 4. Materials and Methods

The study took place at the University Hospital Policlinico of Catania, involving the Laboratory Analysis Unit during a two-year (2021–2023) time interval. Our retrospective analysis included *Aspergillus* spp. isolates isolated from routinely collected respiratory samples such as bronchoalveolar lavage, sputum, and bronchial aspirate. The included samples were derived from intensive care, infectious disease, pneumology, and hematology unit patients. These patients documented probable COVID-19-associated pulmonary aspergillosis (CAPA) according to the current diagnostic guidelines [33]. All the isolates were subjected to a −80 °C storage temperature in sterile water after the conclusion of the routine diagnostic procedures. The strains were restored from these conditions and inoculated into R.P.M.I agar (Liofilchem^®^ s.r.l., Roseto degli Abruzzi, Italy). The MALDI Biotyper^®^ Sirius System (Bruker, Billerica, MA, USA) identified the grown filamentous fungi colonies. The identified *Aspergillus* spp. isolates underwent isavuconazole and amphotericin B susceptibility testing through the MIC Strip method and the gradient test principle [34,35]. All the collected isavuconazole and amphotericin B MIC values were confirmed through the EUCAST broth microdilution guide on filamentous fungi [36]. According to the EUCAST interpretation tables, we considered susceptible all the isolates showing MIC values under or equal to the clinical breakpoint (CBP) along with the isolates reporting MIC values under the epidemiological cut-off (E-COFF). Otherwise, we classified as resistant all the isolates showing MIC values over the CBP along with the ones revealing MIC values under or equal to the E-COFF [14,37]. 

Isavuconazole and amphotericin B strips were subsequently arranged to investigate synergy according to the manufacturer’s instructions [38]. According to these recommendations, both single and synergy tests were undertaken on R.P.M.I. agar (Liofilchem^®^ s.r.l., Roseto degli Abruzzi, Italy), which underwent 24–48 h of incubation at 35 °C. The same instructions also established how to manage the synergy MIC results, calculating the corresponding FIC index and consequently labeling each case as antagonism, indifference, synergy, or additivity. Antagonism occurred when the combination had less effect than the single compounds. Indifference is derived from the absence of interactions between the two compounds in the same plate. Additivity occurred when the combination had a greater effect than the single compounds (showing the addition of the single compound’s effect). Synergy emerged when the combination had a more powerful action than the addition of the two compounds [14,39].

The statistical significance of the *Aspergillus* species and the synergy result were performed using the MedCalc Statistical Software version 17.9.2 (MedCalc Software bvba, Ostend, Belgium; http://www.medcalc.org; 2017), reporting the corresponding *p* values. The χ^2^ and Fisher’s exact test established the categorical variables as percentages. The entire protocol did not involve direct actions on human beings. All the experimental analyses only regarded biological samples and mold isolates. The collected results had no clinical or therapeutical application. On these premises, ethical approvals were not mandatory according to our local legislation. 

## 5. Conclusions

Pulmonary aspergillosis represents a severe public health concern. Unfortunately, diagnostic and therapeutical implications often affect this clinical condition, which has recently been recognized as a complication of respiratory viral infections. Isavuconazole and amphotericin B remain fundamental therapeutical alternatives in the cases of severe *Aspergillus* spp. respiratory infections. The literature data do not document sufficient in vitro synergy studies against *Aspergillus* spp. On this premise, our study evaluated the in vitro synergy of isavuconazole and amphotericin B against *Aspergillus* spp. strains isolated from pulmonary aspergillosis cases. All the tested isolates demonstrated low isavuconazole and amphotericin B MIC values, confirming the previous scientific evidence about these molecules alone. Moreover, the combination of isavuconazole and amphotericin B expressed in vitro synergy or additivity episodes against *A. fumigatus*. Otherwise, *A. niger* and *A. flavus* reported a high antagonism rate, along with the high *A. terreus* indifference percentage. The above-mentioned results enhanced the importance of further investigating in vitro synergic activity for antifungal drugs against molds isolates. Future studies should be planned combining isavuconazole and amphotericin B or echinocandins and isavuconazole against filamentous fungi. Certainly, a larger number of isolates may be included within synergy studies to enrich susceptibility data and scientific evidence in the literature. 

## Figures and Tables

**Figure 1 antibiotics-13-01005-f001:**
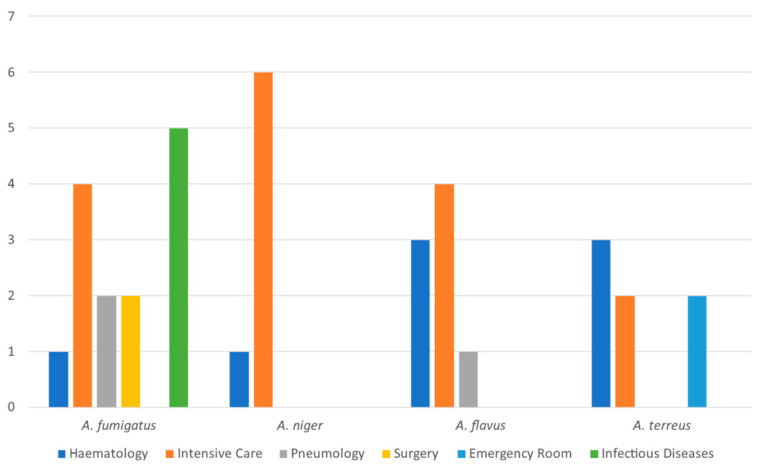
Graphical abstract summarizing the *Aspergillus* species’ distribution within the analyzed hospital units.

**Table 1 antibiotics-13-01005-t001:** Details about isavuconazole and amphotericin B MIC values before the synergy testing for all the investigated isolates.

Species	ISV MIC Strip (mg/L)	AMB B MIC Strip (mg/L)	ISV Synergy MIC Strip (mg/L)	AMB B Synergy MIC Strip (mg/L)
*A. fumigatus*	0.25	0.38	0.25	0.125
*A. fumigatus*	0.19	0.38	0.19	0.25
*A. flavus*	0.25	2	0.19	1
*A. flavus*	0.25	1	0.75	0.125
*A. fumigatus*	1	0.19	0.19	0.75
*A. flavus*	0.064	32	0.032	8
*A. terreus*	0.125	32	0.5	32
*A. fumigatus*	0.25	0.75	0.25	0.25
*A. fumigatus*	0.25	0.25	0.19	0.094
*A. fumigatus*	0.25	0.38	0.125	0.032
*A. fumigatus*	0.38	0.75	0.38	0.25
*A. terreus*	0.094	32	0.094	32
*A. fumigatus*	0.125	0.38	0.064	0.125
*A. fumigatus*	0.047	0.25	0.25	0.75
*A. fumigatus*	0.125	1	0.094	0.25
*A. fumigatus*	0.38	0.50	0.125	0.19
*A. fumigatus*	0.25	0.38	0.125	0.032
*A. terreus*	0.094	32	0.094	32
*A. terreus*	0.002	0.064	0.002	0.064
*A. flavus*	0.094	0.002	0.5	32
*A. niger*	0.002	0.002	0.38	32
*A. terreus*	0.38	32	0.25	8
*A. flavus*	0.012	0.06	0.19	32
*A. flavus*	0.047	0.002	0.25	32
*A. niger*	0.094	3	0.064	1
*A. niger*	0.047	1	0.047	1
*A. niger*	0.002	0.023	0.094	0.75
*A. niger*	0.002	0.002	0.032	0.75
*A. fumigatus*	0.25	0.25	0.19	0.047
*A. fumigatus*	0.19	0.38	0.19	0.25
*A. flavus*	0.25	1	0.75	0.125
*A. flavus*	0.012	0.06	0.19	32
*A. niger*	0.002	0.023	0.094	0.75
*A. niger*	0.002	0.002	0.032	0.75
*A. terreus*	0.094	32	0.094	32
*A. terreus*	0.125	32	0.5	32

Abbreviations: ISV, isavuconazole; AMB B, amphotericin B.

**Table 2 antibiotics-13-01005-t002:** Summary of the collected in vitro susceptibility data about isavuconazole and amphotericin single tests.

*Aspergillus* spp. and Antifungal Agent	EUCAST (E. Def. 9.4)			MIC Strip Test		
	Range (mg/L)	% >ECOFF	S/R	Range (mg/L)	% >ECOFF	S/R
*A. fumigatus* (14)						
Amphotericin B	0.256–1	0	14/0	0.25–1	0	14/0
Isavuconazole	0.016–1	0	14/0	0.047–1	0	14/0
*A. flavus* (8)						
Amphotericin B	0.032->8	12.5	7/1	0.002–32	12.5	7/1
Isavuconazole	0.032–0.512	0	8/0	0.012–0.25	0	8/0
*A. niger* (7)						
Amphotericin B	0.016–2	14.3	6/1	0.002–3	14.3	6/1
Isavuconazole	0.016–0.128	0	7/0	0.002–0.094	0	7/0
*A. terreus* (7)						
Amphotericin B	0.128->8	85.7	1/6	0.064–32	85.7	1/6
Isavuconazole	0.016–0.064	0	7/0	0.002–0.38	0	7/0
TOT (36)						

Abbreviations: S, susceptible; R, resistant. According to the EUCAST interpretation tables, we considered susceptible all the isolates showing MIC values under or equal to the clinical breakpoint (CBP) along with the isolates reporting MIC values under the epidemiological cut-off (E-COFF). Otherwise, we classified as resistant all the isolates showing MIC values over the CBP along with the ones revealing MIC values under or equal to the E-COFF [14].

**Table 3 antibiotics-13-01005-t003:** Synergy evaluation results for all the identified species, along with the corresponding statistical significance.

*Aspergillus* Species	Additivity (%)	Indifference (%)	Antagonism (%)	Synergy (%)	*p* Value
*A. fumigatus* (14)	5 (35.7)	6 (42.8)	1 (7.1)	2 (14.3)	0.020
*A. flavus* (8)	1 (12.5)	3 (37.5)	4 (50)	0	0.599
*A. niger* (7)	1 (14.3)	1 (14.3)	5 (71.4)	0	0.157
*A. terreus* (7)	1 (14.3)	4 (57.1)	2 (28.6)	0	0.674
Total (36)	8 (22.2)	14 (38.9)	12 (33.3)	2 (5.5)	

## Data Availability

All the data gathered during this study have been included within the manuscript.

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
