# Peer review of "Isavuconazole and Amphotericin B Synergic Antifungal Activity: In Vitro Evaluation on Pulmonary Aspergillosis Molds Isolates"

_antibiotics, 2024, doi:10.3390/antibiotics13111005_

Round 1

Reviewer 1 Report

Comments and Suggestions for Authors

1.      Authors concluded that A. fumigatus was the only species to report synergy and the most frequent species to reveal additivity cases in the discussion. Could authors explain why only A. fumigatus showed the synergy effect while other species didn’t.

2.      Line 235: “Isavuconazole confirmed its in vitro effectiveness as a broad-spectrum triazole when compared to the older azole member fluconazole during our experimental analysis”. But I didn’t find the data for fluconazole in the manuscript.

3.      Authors should provide all the data such as MIC of every strain and the MIC value of the groups treated with both isavuconazole and amphotericin B to support the synergic effect.

4.      Line 237, “One A. flavus and one A. niger strains showed a high amphotericin B MIC value,…..” The date in table 1 showed that one A. terreus strain displayed high amphotericin B MIC value (0.064 -32 ).

Author Response

  1. Comment: Authors concluded that fumigatuswas the only species to report synergy and the most frequent species to reveal additivity cases in the discussion. Could authors explain why only A. fumigatus showed the synergy effect while other species didn’t.

            Answer: We hypothesized that A. fumigatus reported synergy and most of the additivity episodes due to the lowest single MIC values gathered for amphotericin B. These data significantly contributed to a favorable FIC index along with low isavuconazole MIC values. We added a sentence about this observation within the discussion section.

  1. Comment: Line 235: “Isavuconazole confirmed its in vitro effectiveness as a broad-spectrum triazole when compared to the older azole member fluconazole during our experimental analysis”. But I didn’t find the data for fluconazole in the manuscript.

            Answer: Thank you for the observation. Fluconazole susceptibility was not included within the study’s aims; thus, we modified the sentence.

  1. Comment: Authors should provide all the data such as MIC of every strain and the MIC value of the groups treated with both isavuconazole and amphotericin B to support the synergic effect.

            Answer: Table 1 has been added to furnish the requested details.

  1. Comment: Line 237, “One  flavusand one A. niger strains showed a high amphotericin B MIC value…” The date in table 1 showed that one A. terreus strain displayed high amphotericin B MIC value (0.064 -32).

            Answer: We are sorry for the typo. Please, find a revised version of Table 1 within the manuscript.

Reviewer 2 Report

Comments and Suggestions for Authors

I suggest correcting table 2 where Aspegillus fumigatus species achieved the highest rates in Indifference and not in Additivity. 

Author Response

Comment: I suggest correcting table 2 where Aspergillus fumigatus species achieved the highest rates in Indifference and not in Additivity. 

Answer: Actually, A. fumigatus reached the highest additivity rate (35.7%) when compared to other species additivity rates (12.5%, 14.3%). Consequently, we confirm the current table 2 version.

Reviewer 3 Report

Comments and Suggestions for Authors

Due to the increasing prevalence of drug resistance and the lack of new drug development, combination therapy using existing drugs has become the prevailing approach. This paper, authored by Calvo and his colleagues, explores a novel combination of two drugs, supported by substantial experimental evidence. The paper is well-written and generally of high quality, but I would like to suggest a few essential improvements for further refinement.

1. Can you explain why pulmonary aspergillosis was chosen as the target among various lung diseases?

2. The connection with COVID-19 disease is unclear. Can you elaborate more on the relationship?

3. To provide contrast, can you include objective data on the efficacy of Isavuconazole and Amphotericin B when administered individually? It doesn’t need to be your own data.

4. Is it possible to provide details such as MIC50 or MIC90?

5. The conclusion is somewhat lacking. Please add a few lines to summarize and encapsulate the key points of the paper.

Author Response

  1. Comment: Can you explain why pulmonary aspergillosis was chosen as the target among various lung diseases?
    Answer: We decided to focus our attention on pulmonary aspergillosis due to its possible diagnostic underestimation and critical therapeutical management. Moreover, our local epidemiology documented an increase in the pulmonary aspergillosis cases since the recent pandemic period. We added some sentences about these considerations within the discussion.
  2. Comment: The connection with COVID-19 disease is unclear. Can you elaborate more on the relationship?
    Answer: We added some sentences within the introduction.
  3. Comment: To provide contrast, can you include objective data on the efficacy of Isavuconazole and Amphotericin B when administered individually? It doesn’t need to be your own data.
    Answer: Some sentenced have been added to the discussion section.
  4. Comment: Is it possible to provide details such as MIC50 or MIC90?
    Answer: Unfortunately, we were not able to calculate these data due to the insufficient strains number for the analysed species. MIC50 and MIC90 are not reliable in the case of small strains number. However, we will furnish similar data in our next studies, aiming to enlarge the strains population for further evaluations.
  5. Comment: The conclusion is somewhat lacking. Please add a few lines to summarize and encapsulate the key points of the paper.
    Answer: Thank you for the observation. The conclusion section has been totally revised.

Round 2

Reviewer 1 Report

Comments and Suggestions for Authors

The revised manuscript is much better and I suggest its acceptance for publication